# CommonKV: Compressing KV Cache with Cross-layer Parameter Sharing

## Abstract

Large Language Models (LLMs) confront significant memory challenges due to the escalating KV cache with increasing sequence length. As a crucial technique, existing cross-layer KV cache sharing methods either necessitate modified model architectures with subsequent pre-training or incur significant performance degradation at high compression rates. To mitigate these challenges, we propose CommonKV, a training-free method for cross-layer KV cache compression through adjacent parameters sharing. Inspired by the high similarity observed in cross-layer hidden states, we utilize Singular Value Decomposition (SVD) to achieve weight sharing across adjacent parameters, resulting in a more easily mergeable latent KV cache. Furthermore, we also introduce an adaptive budget allocation strategy. It dynamically assigns compression budgets based on cosine similarity, ensuring that dissimilar caches are not over-compressed. Experiments across multiple backbone models and benchmarks including LongBench and Ruler demonstrate that the proposed method consistently outperforms existing low-rank and cross-layer approaches at various compression ratios. Moreover, we find that the benefits of CommonKV are orthogonal to other quantization and eviction methods. By integrating these approaches, we can ultimately achieve a 98% compression ratio without significant performance loss.

## 1 Introduction

Large Language Models (Achiam et al., 2023; Yang et al., 2025) have demonstrated significant achievements in long-text understanding and generation tasks (Liu et al., 2025b). They are widely applied in scenarios such as repository-level code generation (Zhang et al., 2024) and complex mathematical reasoning (Chen et al., 2025). As a crucial technique, KV cache stores the keys and values tensor of past contexts in attention module, ensuring the speed of autoregressive generation for LLMs. However, the size of the KV cache is directly proportional to the text length, which imposes a significant memory burden on GPUs (Shi et al., 2024).

To mitigate the issues caused by KV cache during the inference phase, many studies (Zhang et al., 2023; Li et al., 2024a;b) have focused on developing KV cache compression techniques. Leveraging the similarity of information between layers, cross-layer KV cache sharing emerges as a highly promising direction. Existing work (Wu & Tu, 2024; Zuhri et al., 2024; Brandon et al., 2024) primarily focuses on optimizing the attention module structure (Wu et al., 2024) to enable guaranteed cross-layer sharing. YOCO (Sun et al., 2024b) innovatively introduces self-decoder and cross-decoder structures, which significantly accelerate prefilling phase and reduce GPU memory demands by only caching once. In addition to structural enhancements, several studies focus on methods for directly compressing the cross-layer cache of existing LLMs. MiniCache (Liu et al., 2024c) effectively mitigates reconstruction error during cross-layer sharing by disentangling state vectors into magnitude and direction components.

While the above methods considerably alleviate the memory pressure of LLM deployment through cross-layer sharing, several issues remain to be addressed. (1) **Excessive implementation costs.** Most existing methods involve redesigning the Transformer (Vaswani et al., 2017) architecture, which necessitates expensive pre-training to achieve the desired performance from scratch. This makes migrating these techniques to the latest LLMs nearly impossible for reducing KV cache memory. (2) **Performance degradation at high compression rates.** While some methods directly

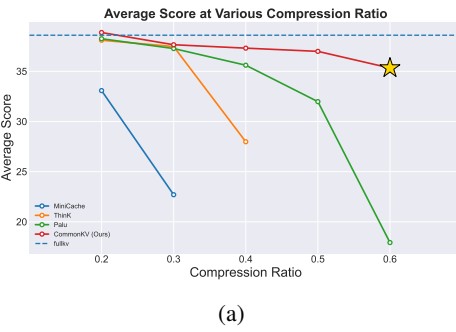 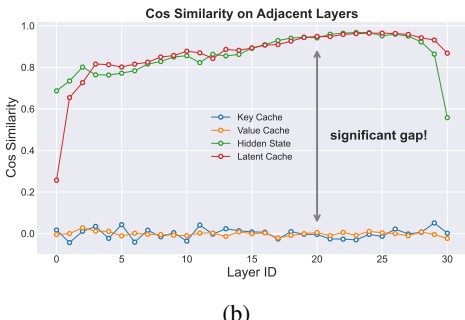

(a)                      (b)

Figure 1: **(a)** Performance of KV cache compression methods under various compression ratios. Proposed CommonKV maintains performance at 0.6 compression ratio, significantly outperforming existing cross-layer sharing and low-rank compression methods. **(b)** Cross-layer cosine similarity of key cache, value cache, hidden state and the proposed latent cache. A significant difference can be observed between the hidden state and the KV cache. This issue can be improved by sharing parameters across layers.

share key-value pairs of existing LLMs, the inherent dissimilarity of KV cache makes it challenging to achieve high compression rates without significant performance degradation. As shown in Figure 1a, the direct sharing method (e.g., MiniCache) suffers from a significant performance drop when the compression ratio exceeds 20%. This suggests the need for a more consistent representation of cross-layer KV cache for compression.

In this paper, we propose **CommonKV**[1], a training-free method that alleviates the aforementioned challenges by merging the cross-layer KV cache through weight parameter sharing. Inspired by the significantly higher similarity of hidden states between adjacent layers compared to the KV cache, we aim to improve the consistency of the KV cache by sharing KV parameter matrices across these layers. Specifically, we obtained partially shared cross-layer weights by concatenating and applying Singular Value Decomposition (SVD) to the KV parameter matrices of adjacent layers. Compared to the original KV cache, the latent cache derived from these shared matrices is more easily merged due to the consistent hidden state input. Furthermore, we propose an adaptive budget allocation strategy to achieve cache sharing with lower performance loss. It dynamically allocates compression budgets across layers based on cosine similarity, preventing performance degradation caused by over-compressing dissimilar caches. Unlike other methods that require training from scratch, CommonKV achieves model conversion with only lightweight, offline SVD, while maintaining LLM performance at high compression rates.

Experiments conducted across various models and benchmarks demonstrate that the proposed method exhibits significant advantages over other sharing and low-rank techniques, particularly at high compression rates. Leveraging latent cache construction and cross-layer sharing, CommonKV maintains over 95% performance on mainstream long-text benchmarks at a 50% compression rate. Specifically, our proposed cross-layer sharing method is orthogonal to existing eviction and quantization techniques. By integrating various compression methods, we can achieve 98% KVCache compression without significant performance loss.

Our Contribution can be summarized as follows:

- We propose CommonKV, a training-free method for KV cache compression. It leverages shared parameters to obtain similar latent cache, boosting performance with cross-layer compression.

- We propose an adaptive budget allocation algorithm that dynamically assigns varying compression rates to regions of differing similarity, mitigating performance degradation from KV cache sharing.

- Extensive experiments validate the effectiveness of the proposed method. Besides, it is orthogonal to other compression algorithms, enabling integration for even higher compression rates.

---

[1]Our code is available at supplementary material.

## 2 OBSERVATIONS

To identify the main challenges in current KV cache sharing methods, we start by statistically analyzing the inter-layer information similarity. We measure the cross-layer cosine similarity of key cache, value cache, and the hidden state input to each layers for Llama3.1-8B-Instruct on the Long-Bench (Bai et al., 2023) benchmark. As shown in Figure 1b, we observe two distinct phenomena:

**(1) Dissimilarity of the adjacent KV cache.** Despite recent efforts (Liu et al., 2024c; Yang et al., 2024b) to compress the KV cache via cross-layer sharing, it is evident from the Figure 1b that the key and value caches in adjacent layers exhibit dissimilarity with respect to cosine similarity. This apparent dissimilarity also explains why such direct merging methods suffer significant performance loss at high compression rates. To enable lower-loss inter-layer sharing, we need to achieve a more consistent representation of key and value information.

**(2) Similarity of the adjacent hidden state.** Additionally, we also compute the similarity between the hidden states input to each layer, which are subsequently transformed into the KV cache through parameter matrices $W_K$ and $W_V$. In contrast to the KV cache, the hidden states across layers exhibit a high degree of consistency in terms of cosine similarity, which aligns with observations of Lee et al. (2024). We hypothesize that this is attributed to the residual connections within the transformer blocks, which also potentially indicates the feasibility of cross-layer cache sharing.

**Motivation.** Given the above observations, we can conclude that the inconsistency of inter-layer KV cache primarily stems from the dissimilarity of the $W_k$ and $W_v$ matrices across different layers. This insight encourages us to explore parameter sharing for key and value matrices across different layers. When similar hidden states are multiplied by the same weight matrices, the resulting latent KV cache would be more consistent, facilitating better cross-layer merging. We first simply validate this idea by sharing matrices across all layers. As shown in Figure 1b, the similarity of the latent cache is significantly improved compared to the KV cache. Moreover, we observe that the cosine similarity of hidden states also varies across layers: lower in shallow layers and higher in deeper ones. This observation motivates the design of an adaptive budget allocation strategy to mitigate the loss introduced by merging.

## 3 METHODOLOGY

In this section, we introduce CommonKV, a cross-layer KV cache sharing approach inspired by the observations discussed above. For an existing large language model, we first perform an offline transformation (§3.1) to enable partial parameter sharing across its layers. Subsequently, we dynamically allocate the compression budget online (§3.2) based on the similarity of the latent KV cache. Some implementation details will be elaborated in §3.3.

### 3.1 CROSS-LAYER PARAMETER SHARING

**Problem Statement.** Existing LLMs primarily employ Multi-Head Attention (MHA) (Vaswani et al., 2017) or Grouped-Query Attention (GQA) (Ainslie et al., 2023) as the main structures for their attention modules. The acquisition of the KV cache can be formalized as:

$$k_i^l = x_i^l W_k^l, \quad v_i^l = x_i^l W_v^l \tag{1}$$

Where $W_k^l$ and $W_v^l$ are the weight matrices corresponding to the l-th layer, $x_i^l$ is the hidden state of $token_i$ at the l-th layer, and $k_i^l, v_i^l$ denotes the corresponding cache.

As discussed in Section 2, given the inherent similarity between $x_i^l$ and $x_i^{l+1}$ across layers, the dissimilarity in cross-layer KV pairs primarily stems from the differences in their respective weight matrices. Therefore, a straightforward approach is to share the KV matrices across layers, aiming to achieve more similar projected caches.

**Cross-layer parameter sharing.** Inspired by related work (Wang et al., 2024) in model compression, we employ a cross-layer concatenated SVD method for decomposing the weight matrices. As

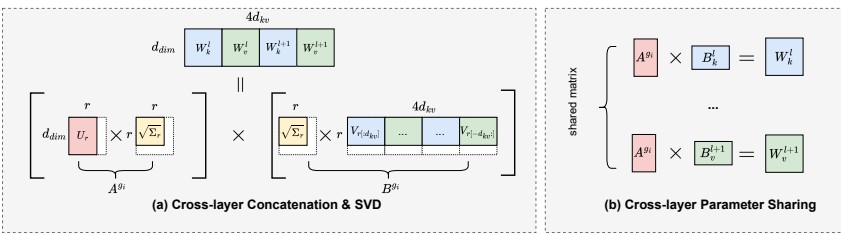

Figure 2: An Illustration of Cross-Layer Weight SVD Decomposition. Through parameter sharing, we obtain a shared matrix $A_{g_i}$ and layer-specific matrices $B_{k/v}^l$ for each layer.

shown in Figure 2, we first concatenate the KV matrices of adjacent layer groups $g_i$, which can be defined as:

$$W_{g_i} = \left[ W_k^l; W_v^l; ...; W_k^{l+|g_i|}; W_v^{l+|g_i|} \right] \tag{2}$$

Subsequently, SVD decomposition is applied to the concatenated matrix $W_{g_i}$ to derive the corresponding shared and layer-specific parameters:

$$W_{g_i} \approx U_r \Sigma_r V_r^T = \left[ U_r \sqrt{\Sigma_r} \right] \left[ \sqrt{\Sigma_r} V_r^T \right]$$
$$= [A^{g_i}] \left[ B_k^l; B_v^l; ...; B_k^{l+|g_i|}; B_v^{l+|g_i|} \right] \tag{3}$$

Where $r$ denotes the chosen SVD rank, $A^{g_i}$ represents the shared parameters for Group i, and $B_{k/v}^l$ refers to the right part matrix of SVD sliced according to the order in Equation 2. Finally, we can obtain an approximate representation of the original parameter matrix, consisting of the product of a shared matrix and a specific matrix:

$$W_k^{l+|g_i|} \approx A^{g_i} B_k^{l+|g_i|}, \quad W_v^{l+|g_i|} \approx A^{g_i} B_v^{l+|g_i|} \tag{4}$$

**Consistent latent KV Cache.** By leveraging the shared parameter matrix, the original KV cache construction can be expressed as:

$$k_i^l = x_i^l A^{g_i} B_k^l, \quad v_i^l = x_i^l A^{g_i} B_v^l \tag{5}$$

Instead of caching the original key and value vectors, we choose to cache the more consistent latent KV representations:

$$h_i^l = x_i^l A^{g_i} \tag{6}$$

After pre-filling, we perform cross-layer merging on the obtained latent cache $h_i^l$ through averaging, achieving KV cache capacity compression. During the inference phase, we restore the latent KV cache to the original cache for attention computation with RoPE (Su et al., 2024) position embeddings, which is somewhat similar to the process of Multi-head Latent Attention (MLA) (Liu et al., 2024b).

## 3.2 Adaptive Budget Allocation

**Group score calculation.** While parameter sharing yields a more consistent latent KV cache, we still observe that different groups exhibit varying sensitivities to merging during inference. Compared to deeper layers, shallower layers of the cache typically show greater inter-layer differences, which can lead to significant performance degradation upon merging. This led us to allocate different budgets across groups to ensure the effectiveness of CommonKV.

In practice, we use the cosine similarity between latent cache layers as an evaluation metric to guide dynamic budget allocation during the inference phase. The score for group $g_i$ can be formulated as:

$$\text{score}(g_i) = \frac{\sum_{t \in g_i} \text{sim}(h_t^l, h_t^{l+|g_i|})}{|g_i|} \tag{7}$$

where $t$ represents tokens in group $g_i$ across layers. To reduce computational overhead, we only use the cosine similarity between the first and last layers within each group as its score. After calculating the score of each group, we selectively merge groups based on the current compression ratio. For instance, given a group size of 4 and a desired compression ratio of 0.5, we merge and share the top 50% most similar groups to meet the global compression target.

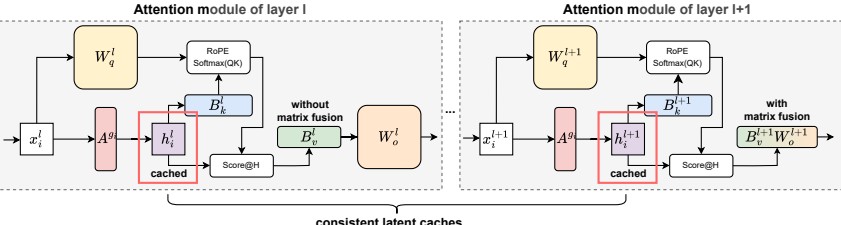

Figure 3: The main architectural diagram of the proposed CommonKV. Through cross-layer parameter sharing, we can use shared matrices $A^{g_i}$ to obtain the latent KV cache $h_i^l$ for each layer. Additionally, we further reduce inference computation by fusing matrices $B_v^l$ and $W_o^l$.

**Fisher information-based merging.** After obtaining the indices of the layers to be merged based on the budget, we perform latent sharing through a weighted merge. Assuming the cache derived from more informative weights is more sensitive, we propose using the Fisher information (Ly et al., 2017; Liu et al., 2021) of the corresponding layers as the merging weights. This can be formulated as:

$$h^{g_i} = \frac{1}{C} \sum_{l \in g_i} (\mathcal{F}(W_k^l) + \mathcal{F}(W_v^l)) h_i^l \tag{8}$$

where $C$ is a normalization constant, and $\mathcal{F}(\cdot)$ denotes the Fisher information of the corresponding weight parameters. Given that the key and value for each layer share a common latent cache, we use the sum of their Fisher information as weights.

### 3.3 MERGING AND RESTORATION DETAILS

After performing offline parameter sharing on an existing LLM, we leverage the similarity of the latent KV cache during inference to achieve inter-layer compression. To balance performance and speed during the inference phase, we implement some specific designs for merging and restoration.

**Position embeddings.** Compared to original key cache, the latent cache $h_t^l$ employing by CommonKV does not contain any positional information. This necessitates recalculating the corresponding position embeddings based on the position ids during each decoding step. To further reduce computation time, we pre-calculate the position embeddings shared across all layers before the first layer's computation.

**Cache merging strategies.** To ensure generation quality, we follow (Chang et al., 2025)'s setting and only compress the KV cache conducted during prefilling phase. Although the KV cache for the output segment typically accounts for a small proportion, for a fair comparison, we calculate the average compression ratio of both the compressed and uncompressed parts as the final compression rate.

**Matrix fusion.** As illustrated in Figure 3, while compressing KV cache memory, the proposed CommonKV introduces some computational overhead. During the reconstruction phase, an additional two reconstruction KV matrices need to be computed, which impacts inference performance. To alleviate this issue, we offline-merged the $B$ matrix with the normally computed $W_o$ matrix.

$$\begin{aligned} O &= \text{softmax}(\frac{QK^T}{\sqrt{d_k}})VW_o \\ &= \text{softmax}(\frac{QK^T}{\sqrt{d_k}})XA^{g_i}\left[B_v^{l+|g_i|}W_o\right] \end{aligned} \tag{9}$$

This matrix fusion operation can effectively reduce the computational cost of reconstructing the KV cache. Moreover, we provide a detailed discussion of inference latency for different methods in Section 5.2.

## 4 EXPERIMENTS

### 4.1 EXPERIMENTAL SETUP

**Evaluation Datasets.** We evaluate CommonKV using two widely adopted long-context datasets: LongBench (Bai et al., 2023) and Ruler (Hsieh et al., 2024). These datasets represent real-world and synthetic data, respectively. The context window for all models was set to 8k. For Ruler, we follow the setup from Chang et al. (2025) and report the average performance across the corresponding multiple tasks. The compression ratio is defined as $1 - \frac{\text{Compressed KV}}{\text{Original KV}}$. A higher ratio indicates greater memory savings.

**Selected Baselines.** The proposed CommonKV method, which incorporates SVD, can be considered a hybrid of low-rank and cross-layer sharing methods. Therefore, we primarily select several mainstream low-rank and cross-layer methods as baselines for comparison:

- **MiniCache** (Liu et al., 2024c) employs a disentangled representation of the KV cache to achieve training-free cross-layer sharing.
- **ThinK** (Xu et al., 2024) achieves key cache compression and efficient QK computation by pruning the unnecessary dimensions of the key vectors.
- **Palu** (Chang et al., 2024) decomposes the weight matrices of each layer, thereby caching a lower-rank hidden KV cache.

Additionally, xKV (Chang et al., 2025) achieves good compression by extracting the common information from the inter-layer cache via online SVD decomposition. However, its significant time overhead compared to other methods warrants a separate discussion in Section 5.2. We compare all methods at compression rates ranging from 0.3 to 0.6. Beyond a compression rate of 0.6, all methods experience a significant performance degradation.

**Implementation details.** We conduct main experiments on two mainstream LLMs: Llama3.1-8B-Instruct (Dubey et al., 2024) and Mistral-v0.2-7B-Instruct (Jiang et al., 2023). For these GQA-based models, we choose a group size of 4 for KV cache sharing. To achieve various compression ratios while optimally preserving performance, the SVD rank is set to $0.7 \cdot d_{\text{hidden}}$ for compression ratios of 0.3 and 0.5, and to $0.6 \cdot d_{\text{hidden}}$ for a ratio of 0.6. Following setup of Chang et al. (2024), we utilize 2048 samples with a sequence length of 1024 from Wikitext-2 (Merity et al., 2016) to compute the Fisher Information during the merging phase.

**Challenges of GQA.** For traditional MHA architectures, sharing the KV cache every N layers can theoretically achieve a compression ratio of 1/N. Unfortunately, for GQA models, $d_{kv}$ is usually smaller than $d_{hidden}$. This means that after cross-layer SVD, latent cache of a single layer must be slightly larger than the original cache to maintain performance. To alleviate this issue, we first concatenate and share the Key and Value matrices, which increases the rank of the parameters for each layer. Additionally, during the SVD process, we reduce the rank of the concatenated matrix to further compress the storage of its latent cache. Ultimately, we achieved a 0.5 compression ratio on mainstream GQA models by merging every 4 layers, with the flexibility to adjust the SVD rank.

### 4.2 MAIN RESULTS

**Overall performance.** We benchmark the performance of CommonKV against other comparable strong baselines on the mainstream long-context tasks. As shown in Table 1, the proposed method demonstrates significant advantages across multiple models and various compression ratios. Specifically, CommonKV can maintain 95% end-to-end performance at a 0.5 compression ratio, effectively alleviating the memory pressure caused by KV cache in long-context scenarios. This is primarily attributable to the more consistent latent KV cache resulting from cross-layer parameter sharing, which further elevates the ceiling of training-free cross-layer sharing methods.

**Compared with cross-layer methods.** As discussed in Section 2, the dissimilarity of the KV cache across layers is the main challenge for existing cross-layer sharing methods. Despite employing magnitude and direction vector extraction for the KV cache of adjacent layers, MiniCache

| Ratio | Method | LongBench (Bai et al., 2023) | | | | | Ruler (Hsieh et al., 2024) | | | | | | | Avg. |
|---|---|---|---|---|---|---|---|---|---|---|---|---|---|---|
| | | QA | Sum. | FShot | Synth. | Code | NS | NMK | NMQ | NMV | QA | VT | FWE | |
| | | | | | | *Llama3.1-8B-Instruct* | | | | | | | | |
| 0.0 | Full KV | 17.20 | 27.85 | 68.31 | 39.87 | 61.15 | 100.00 | 100.00 | 100.00 | 95.80 | 76.76 | 99.90 | 95.90 | 73.55 |
| | MiniCache | 7.60 | 18.19 | 51.17 | 15.01 | 36.64 | 11.50 | 1.10 | 0.60 | 0.60 | 43.75 | 23.80 | 28.50 | 18.90 |
| 0.3 | ThinK | 16.52 | 27.13 | 67.00 | 39.34 | 58.28 | 100.00 | 99.90 | 99.70 | 87.75 | 74.39 | 99.88 | 96.60 | **72.34** |
| | Palu | 18.76 | 27.55 | 67.40 | 36.21 | 54.97 | 99.10 | 99.90 | 99.60 | 91.75 | 72.96 | 99.96 | 95.40 | 72.14 |
| | CommonKV | 17.12 | 25.38 | 66.54 | 41.48 | 58.58 | 99.80 | 99.00 | 99.90 | 96.65 | 71.59 | 98.88 | 94.90 | 72.31 |
| | ThinK* | 7.97 | 17.34 | 50.19 | 37.84 | 46.58 | 1.90 | 1.90 | 0.55 | 1.10 | 51.14 | 1.52 | 16.13 | 18.57 |
| 0.5 | Palu | 17.87 | 26.27 | 65.97 | 25.45 | 38.43 | 98.80 | 98.60 | 98.60 | 94.20 | 65.07 | 98.00 | 93.07 | 68.79 |
| | CommonKV | 16.74 | 24.59 | 64.62 | 41.40 | 57.86 | 98.70 | 97.20 | 99.40 | 95.20 | 71.66 | 99.20 | 94.60 | **71.59** |
| 0.6 | Palu | 9.20 | 16.31 | 46.95 | 2.62 | 23.32 | 99.60 | 86.50 | 55.25 | 45.15 | 39.16 | 84.88 | 69.00 | 50.59 |
| | CommonKV | 15.74 | 23.90 | 61.77 | 41.30 | 53.54 | 98.90 | 94.40 | 98.50 | 93.05 | 58.34 | 95.40 | 88.80 | **68.19** |
| | | | | | | *Mistral-v0.2-7B-Instruct* | | | | | | | | |
| 0.0 | Full KV | 32.30 | 27.56 | 64.77 | 37.13 | 55.17 | 99.90 | 98.70 | 99.00 | 96.50 | 68.04 | 99.32 | 91.73 | 73.07 |
| | MiniCache | 14.80 | 19.86 | 45.14 | 2.46 | 34.79 | 31.80 | 10.90 | 5.50 | 5.50 | 35.97 | 25.21 | 39.93 | 22.83 |
| 0.3 | ThinK | 31.59 | 26.93 | 63.30 | 27.55 | 51.39 | 100.00 | 97.80 | 95.80 | 84.75 | 65.59 | 99.30 | 91.60 | 70.72 |
| | Palu | 31.10 | 27.04 | 64.40 | 35.30 | 50.50 | 99.90 | 95.50 | 95.05 | 93.50 | 66.41 | 99.20 | 91.40 | 71.39 |
| | CommonKV | 31.74 | 25.81 | 63.26 | 35.04 | 55.15 | 99.70 | 97.40 | 97.65 | 95.35 | 65.01 | 98.28 | 91.20 | **71.84** |
| | ThinK* | 26.79 | 20.26 | 52.92 | 23.44 | 42.59 | 58.50 | 41.30 | 29.10 | 27.85 | 54.63 | 21.90 | 22.80 | 37.71 |
| 0.5 | Palu | 30.70 | 26.31 | 63.25 | 22.01 | 45.42 | 99.80 | 89.30 | 94.00 | 92.60 | 65.46 | 95.70 | 81.50 | 68.21 |
| | CommonKV | 30.41 | 24.36 | 59.62 | 33.45 | 53.35 | 99.60 | 95.60 | 94.65 | 94.20 | 64.79 | 97.88 | 90.83 | **70.57** |
| 0.6 | Palu | 27.46 | 25.19 | 61.49 | 12.81 | 31.94 | 99.80 | 81.75 | 94.85 | 83.75 | 59.81 | 88.30 | 73.17 | 63.07 |
| | CommonKV | 28.56 | 23.84 | 59.91 | 28.85 | 54.28 | 99.50 | 94.30 | 92.65 | 93.00 | 61.29 | 93.20 | 84.80 | **68.61** |

Table 1: Compression performance comparison on long-context datasets. **Bold** indicates the best performance at an equivalent compression ratio. Due to significant performance loss, some methods do not report results for all compression ratios. Since Think only compresses the key cache, we calculate the final compression ratio as the average of the key-vale pairs. ∗ denotes an actual compression ratio of 0.4.

suffers significantly from this inherent dissimilarity, leading to substantial performance degradation even at a mere 0.2 compression ratio. In contrast, the proposed method fundamentally mitigates the dissimilarity issue by sharing parameters across layers, naturally reducing the loss from merging. Under Fisher information-based weighted merging, the CommonKV achieves dynamic four-layer merging, exhibiting no significant performance degradation even at a 0.6 compression ratio.

**Compared with low-rank methods.** Additionally, considering CommonKV leverages the concept of SVD decomposition for compression, we also compare it with several low-rank KV cache compression methods. Compared to traditional cross-layer merging methods, low-rank methods perform better at lower compression ratios, indicating that existing KV cache indeed has some redundancy in their dimensions. However, low-rank methods have a clear upper limit on their compression ratio, with a noticeable decline observable after exceeding 0.5 compression. Halving the dimensionality while maintaining performance is far more challenging than merging two KV cache layers, due to the inherent inter-layer connections. The proposed CommonKV greatly extends the upper bound of low-rank methods through cross-layer merging, demonstrating a clear advantage over the powerful baseline method ThinK and Palu at most compression ratios.

## 5 ANALYSIS

### 5.1 EFFECT OF MERGING MECHANISM

To validate the effectiveness of our proposed Adaptive budget allocation strategy, we conduct an experimental analysis of the merging mechanism on LongBench, using Llama3.1-8B-Instruct. All merging strategies are built upon the proposed CommonKV method, tailored for a more consistent latent KV cache, with a selected compression ratio of 0.5. We compared two allocation strategies: static budgeting and dynamic budgeting, encompassing

| Method | Adapt. | QA | Sum. | FShot | Synth. | Code | Avg. |
|---|---|---|---|---|---|---|---|
| Full KV | - | 17.20 | 27.85 | 68.31 | 39.87 | 61.15 | 38.60 |
| MCache | ✗ | 12.74 | 23.06 | 60.33 | 41.00 | 55.08 | 34.16 |
| Mean | ✗ | 13.52 | 23.36 | 60.63 | 40.66 | 52.96 | 34.11 |
| Fisher | ✗ | 12.82 | 23.52 | 63.48 | 40.30 | 52.20 | 34.19 |
| Shallow | ✓ | 16.00 | 25.02 | 68.44 | 35.50 | 57.72 | 36.45 |
| Deep | ✓ | 14.56 | 23.80 | 65.57 | 35.50 | 57.39 | 35.23 |
| Mean | ✓ | 16.20 | 24.54 | 66.84 | 38.50 | 57.03 | 36.55 |
| Fisher | ✓ | 16.74 | 24.59 | 64.62 | 41.40 | 57.86 | **36.99** |

Table 2: Performance of different merging mechanisms on LongBench at a compression ratio of 0.5. **Adapt.** indicates whether dynamic budget allocation is used.

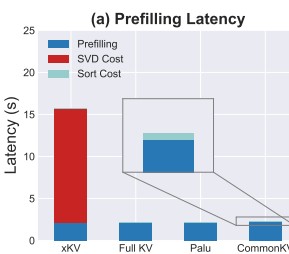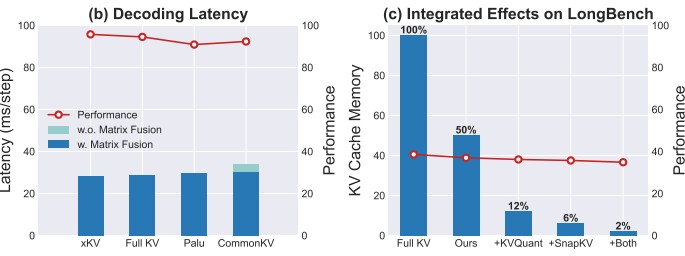

Figure 4: Inference Latency and Integration Analysis of CommonKV on Llama3.1-8B-Instruct. **(a)** Average latency of different methods during the prefilling stage. SVD Cost refers to the online SVD decomposition latency of xKV, while Sort Cost denotes the adaptive budget allocation overhead of CommonKV. **(b)** Average latency of different methods during the decoding stage. **(c)** Memory footprint of the KV cache integrated with different methods and performance on downstream tasks.

several distinct merging methods: **Mean** serves as the most straightforward baseline, representing a simple average. **MCache** represents the application of MiniCache (Liu et al., 2024c) to the latent KV cache. **Shallow** and **Deep** represent two extreme cases where only the latent cache of the first or last layer, respectively, is retained as a representative. **Fisher** is the proposed Fisher information based merging method, which merges the KV cache of adjacent layers based on the Fisher information of the parameters of corresponding layers. For consistent compression ratios, we use the full rank for static methods to achieve global compression, while dynamic methods employ a lower rank for group-adaptive compression.

The experimental results, as shown in Table 2, indicate that dynamic methods still possess a significant advantage over static methods, even when employing a lower-rank compression. Using a weighted averaging approach can better preserve the performance on downstream tasks compared to sharing the cache of specific layers. We can also see that the proposed method does not conflict with existing merging methods like MiniCache. The consistent latent cache can further raise the upper bound of current algorithms.

## 5.2 INFERENCE LATENCY ANALYSIS

Additionally, due to the introduction of an extra restoration operation, we also analyze the inference latency of the proposed method. Specifically, we compared the prefill and decoding latency of our method against the similar methods Palu (Chang et al., 2024) and xKV (Chang et al., 2025) at an 8K context length. The latency test is conducted on a single RTX A6000 GPU.

As shown in Figure 4(a), the online SVD of the cache introduces non-negligible latency at an 8K context length. The SVD construction time alone accounts for more than six times the entire prefilling latency, whereas our cost for constructing the dynamic budget is only 5%. In terms of decoding latency, as shown in Figure 4(b), the proposed method exhibits no significant disadvantage compared to normal auto-regressive decoding after optimization of the matrix merging. Additionally, with nearly a 6x reduction in latency, CommonKV achieves comparable performance to both the online method xKV and the original baseline at a 0.5 compression ratio on Ruler benchmark.

## 5.3 INTEGRATION WITH OTHER METHODS

As a relatively independent compression method, cross-layer merging has the potential to integrate with other KV cache compression techniques to further enhance compression ratios. We validate the orthogonality of CommonKV with other methods on LongBench. Specifically, we select SnapKV (Li et al., 2024b) to represent eviction methods and KVQuant (Hooper et al., 2024) to represent quantization methods. For quantization, we directly apply K4V4 quantization to the shared latent KV cache. Regarding the eviction strategy, we opt to share eviction indices across merged layers, applying them to the sharing latent KV cache.

The experimental results are shown in Figure 4(c). The cross-layer sharing method shows strong orthogonality with the other two categories of methods. By integrating these approaches, we can achieve 98% KV cache compression. Additionally, we observe a phenomenon similar to Yang et al.

(2024a), where using a single set of eviction indices across adjacent layers resulted in a much smaller performance loss than anticipated. This further validates the inherent similarity between layers.

# 6 RELATED WORK

In long-context tasks, the memory footprint of the KV cache has reached the same order of magnitude as the model itself. Consequently, a growing number of studies (Liu et al., 2024d; Li et al., 2024b; Hooper et al., 2024) are dedicated to KV cache compression. The proposed CommonKV combines the principles of low-rank approximation and cross-layer merging. Recently, many efforts have also focused on KV cache compression from these two perspectives.

## 6.1 CROSS-LAYER COMPRESSION

**Architecture.** To mitigate the KV cache problem in traditional Transformer (Vaswani et al., 2017) architectures, some studies (Wu & Tu, 2024) have focused on optimizing the model architecture to enable cross-layer cache sharing. YOCO (Sun et al., 2024b) partitions all blocks into two parts: the self-decoder and the cross-decoder. which retains the KV cache only from the self-decoder portion. Similarly, both MLKV (Zuhri et al., 2024) and CLA (Brandon et al., 2024) employ a Cross-Attention design to enable cross-layer KV cache sharing, thereby reducing deployment overhead. CMLA (Yang et al., 2024c) applies a sharing technique to the MLA model (Liu et al., 2024b). When combined with continuous training, it compresses the KV cache to 2% through full-layer sharing.

**Algorithm.** Several plug-and-play algorithms have also been proposed to merge the inter-layer KV cache. MiniCache (Liu et al., 2024c) leverages spherical interpolation to achieve efficient inter-layer cache merging and reconstruction. While xKV (Chang et al., 2025) explores the inherent similarity of inter-layer caches from the perspective of Centered Kernel Alignment (CKA). In addition to similarity, KVSharer (Yang et al., 2024b) proposes a counterintuitive strategy for cross-layer sharing, demonstrating that sharing dissimilar KV caches better preserves model performance than similarity-based approaches.

## 6.2 LOW-RANK COMPRESSION

**Architecture.** Traditional MHA (Vaswani et al., 2017) is designed with a one-to-one correspondence between each query and its key-value pair, which results in dimensional redundancy. Multi-Query Attention (MQA) (Shazeer, 2019) and Grouped-Query Attention (GQA) (Ainslie et al., 2023) have demonstrated that a single group of queries can still perform well with a single KV pair, which significantly reduces the dimension of the KV cache. Multi-head Latent Attention (MLA) (Liu et al., 2024a;b) restores a richer KV cache representation during inference by storing a low-rank KV cache.

**Algorithm.** Shadow KV (Sun et al., 2024a) uses landmarks to reconstruct the SVD-compressed KV cache. It significantly improves system throughput combined with offloading operations. Furthermore, xKV (Chang et al., 2025) explores the potential for cross-layer SVD compression. By leveraging the inherent similarity between layers, it achieves a higher compression ratio. By operating on position-related head dimensions, FourierAttention (Liu et al., 2025a) performs fixed-length compression on context-insensitive dimensions, leading to a notable reduction in memory usage.

# 7 CONCLUSION

In this paper, we introduce CommonKV, a training-free KV cache cross-layer compression method. By exploiting the similarity of hidden states across layers, it achieves effective cross-layer latent cache merging through inter-layer parameter sharing. Extensive experiments demonstrate that our proposed method significantly outperforms existing sharing and low-rank approaches. Furthermore, our method is orthogonal to various existing KV cache compression techniques, which highlights its broad applicability.

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

## A  AI TOOLS

In this manuscript, AI tools, including ChatGPT, were employed only to improve the wording and clarity of the text, without influencing the content or ideas presented.

