# OpenReview forum: "CommonKV: Compressing KV Cache with Cross-layer Parameter Sharing"
_ICLR.cc/2026/Conference — ICLR 2026 Conference Withdrawn Submission_

### Official Review · Reviewer_kcrW · 2025-10-31

**Soundness:** 3
**Presentation:** 3
**Contribution:** 2
**Rating:** 4
**Confidence:** 2

**Summary:**

The authors note the similarity in hidden representation across layers -- however, there is a high dissimilarity in crosslayer KV pairs due to differences in W_k and W_v. To alleviate KV-footprint growth, concatenate adjacent layer weight matrices W_k, W_v and run SVD, leading to a mergeable latent KV cache,. Further, the authors use cosine similarity to adapt the budget allocation. This is a training-free method with RoPE reapplied upon reconstructions.

**Strengths:**

- an interesting observation regarding the KV-generator being compressible is utilized to effectively reduce inference overhead.
- no pretraining/fine-tuning makes it quite practical
- strong performance with good  latency analysis. also orthogonal to other optimizations (eviction/quantization)
- discusses SVD cost of xKV well, does not have the same limitations, addresses a good set of baselines.

**Weaknesses:**

- Several figure references are broken, TODO -- should be removed.

**Questions:**

- Comments on comparison with / Applicability to MLA would be useful
- Online SVD becomes less than 10% of prefill at 128K, but this approach requires significant changes at deployment (reparametrize attention, changes for latent cache, rope, etc.) -- is it trivial to switch between this method and vanilla attention at inference? i.e., at lower sequence lengths can we do vanilla attention where prefill cost is high? What is the prefill cost at lower sequence length, and the overhead as the conversation starts at low token count but increases over multi-turn?

---

> ### Author Response · Authors · 2025-11-25
> **Response to Reviewer kcrW**
>
> Thank you for your constructive and valuable suggestions. We provide our responses as follows:
>
> ### **1. Correction of formatting issues (corresponding to Weakness 1).**
>
> We apologize for the inconvenience caused during your review due to our oversight. Several typographical errors are already corrected in the latest version. We appreciate your careful review and meaningful suggestions.
>
> ### **2. Adaptability to MLA models (corresponding to Question 1).**
>
> This is a highly meaningful question.
> Methodologically, CommonKV can be adapted to models with an MLA architecture.
> However, similar to the challenges faced with the GQA architecture, such KV cache compression methods can be viewed as trainable low-rank compressions of the KV cache. These architectures reduce the potential gains achievable by existing low-rank KV cache compression techniques.
> As a result, existing purely low-rank methods cannot achieve further compression on GQA and MLA architectures. In contrast, CommonKV can further reduce the KV cache size through cross-layer merging, allowing more relaxed rank settings for SVD at the same compression ratio.
>
> ### **3. Flexible adaptations for practical deployment (corresponding to Question 2).**
>
> This is indeed a point worth exploring. For the proposed CommonKV, it may be necessary to compute the inverse of each layer’s specific matrix—either through trainable methods or numerical solutions—to enable a conversion between vanilla attention and the proposed approach.
> For short texts, we still use standard attention to obtain the conventional KV cache. Once the length reaches a certain threshold, we convert the KV cache into the latent cache described in the paper using the inverse matrices for merging. This may represent an interesting direction for future work.

---

> > ### Author Response · Authors · 2025-11-28
> > **Look Forward to More Feedback**
> >
> > Dear Reviewer kcrW,
> >
> > We sincerely appreciate your thoughtful and constructive feedback. We have diligently addressed each of your concerns in our point-by-point rebuttal.
> >
> > We hope our response has alleviated your concerns and illuminated the value of our work. If so, we would be immensely grateful if you could reconsider your recommendation. We assure you that all your insightful comments will be meticulously incorporated into the final manuscript. We have earnestly integrated feedback from all four reviewers and hope this is evident in your evaluation.
> >
> > Thank you for dedicating your valuable time to review our response.
> >
> > With deepest gratitude,
> >
> > Paper 16049 Authors

---

### Official Review · Reviewer_27QT · 2025-10-31

**Soundness:** 3
**Presentation:** 2
**Contribution:** 2
**Rating:** 6
**Confidence:** 3

**Summary:**

* The paper proposes an SVD-based method to enable parameter sharing across KV Cache layers along with an adaptive budget strategy.  Experiments across multiple models and benchmarks show up to 98% compression ratio without significant performance loss.

**Strengths:**

* Neat insight into using parameter sharing across layers to compress based on similarity between adjacent layers of the KV Cache.
* Allows dynamic adaptation across different groups of KV layers for better performance.
* Training-free method of KV merging leads to lower offline compute overhead
* Results are very impressive compared to all the baselines.

**Weaknesses:**

* *To reduce computational overhead, we only use the cosine similarity between the first and last layers within each group as its score* It needs further justification that it is sufficient to use just the first and last layer within each group without sacrificing quality.
* It is not clear why the latent cache is more easily mergeable. I see the claim that it has more consistent hidden states, but this claim needs further explanation. This is especially important as there is a two-way overhead of constructing the latent cache and reconstructing the original KV cache. Secondly, it is not clear why parameter sharing based on just "cosine similarity" would not degrade quality.
* NIT: References to Figures are broken in the text.

**Questions:**

* Table 1 says *Due to significant performance loss, some methods do not report results for all compression ratios*. I think it would still be useful to see the comparison as even the case which is included (MiniCache for CR 0.3, the performance loss is very significant)

---

> ### Author Response · Authors · 2025-11-25
> **Response to Reviewer 27QT**
>
> Thank you for your constructive and valuable suggestions. We provide our responses as follows:
>
> ### **1. Computation of similarity across different groups (corresponding to Weaknesses1).**
>
> The similarity within a group is primarily used to determine which layers to prioritize for compression under a given budget.
> In the paper, we heuristically compute the similarity using the first and last layers of each group, as this pair represents the combination with the largest difference within the group after KV cache sharing, providing a reasonably representative measure.
> To address reviewers’ concerns, we also conduct experiments using the average pairwise cosine similarity within each group as the metric, with results shown below.
>
> | Method               | QA     | SUM    | FShot  | Synth  | Code   | Avg    |
> |---------------------|--------|--------|--------|--------|--------|--------|
> | baseline             | 17.20  | 27.85  | 68.31  | 39.87  | 61.15  | 42.88  |
> | adjacent layers avg  | 19.69  | 24.62  | 64.71  | 41.42  | 57.84  | 41.66  |
> | shallow & deep layers only| 19.88  | 24.59  | 64.62  | 41.40  | 57.87  | 41.67  |
>
> As shown in the figure above, using the full average similarity versus the similarity between the first and last layers results in negligible differences in performance. To reduce additional computation during inference, we ultimately choose to compute the similarity only once between the first and last layers to rank the groups.
>
> ### **2. Discussion of the latent cache (corresponding to Weakness 2).**
>
> We apologize for the confusion. As shown in Figure 1b, after parameter sharing, the similarity of the latent cache is substantially higher than that of the KV cache.
> This similarity arises from the phenomenon of similar hidden states, as well as the fact that both are multiplied by the same shared matrix.
> Different layers use distinct matrices when reconstructing the original KV cache to preserve each layer’s specificity; only the latent cache, as an intermediate result, is shared.
>
> For the second question, the heuristic perspective is that if the caches within a group exhibit high similarity, the loss incurred by sharing and merging these caches is minimal.
> Parameter sharing itself is used solely to obtain more similar latent caches and does not significantly affect performance.
> The cross-layer sharing of the latent cache is the key operation that influences the final performance.
>
> ### **3. Correction of formatting issues (corresponding to Weakness 3).**
>
> We apologize for the inconvenience caused during your review due to our oversight. Several typographical errors are already corrected in the latest version. We appreciate your careful review and meaningful suggestions.
>
> ### **4. Selection of main experimental settings (corresponding to Question 1).**
>
> This is indeed a very good suggestion.
> Considering that these methods collapse in performance under high compression ratios (e.g., failing to generate coherent responses) and the space limitations of the main table, we do not report these results in the primary table.
> We will consider including the complete results in the appendix in future revisions.
> Thank you for your suggestion.

---

### Official Review · Reviewer_F8ju · 2025-11-01

**Soundness:** 3
**Presentation:** 3
**Contribution:** 3
**Rating:** 6
**Confidence:** 3

**Summary:**

This paper proposes CommonKV, a training-free KV cache compression method. It addresses the dissimilarity of KV caches across layers by using offline Singular Value Decomposition (SVD) to share parameters between adjacent layers. This creates a highly consistent "latent KV cache" from similar hidden states, which can be merged with significantly less performance loss than merging the original KV caches directly.

**Strengths:**

The method's core strength is its novel idea of creating a more consistent latent cache via parameter sharing, which directly addresses the root cause of poor performance in direct KV cache merging. Its training-free nature makes it highly practical and easy to apply to existing models. CommonKV demonstrates superior empirical performance over baselines at high compression ratios with minimal inference latency overhead. Furthermore, its ability to be combined with quantization and eviction methods to achieve up to 98% compression is a significant advantage .

**Weaknesses:**

The paper would be stronger with a more detailed sensitivity analysis of the SVD rank hyperparameter. Additionally, the handling of GQA models feels like a workaround, and a deeper analysis of the architectural interaction would be beneficial.

**Questions:**

The idea of low rank decomposition and weight sharing sounds very related to "LORC: Low-Rank Compression for LLMs KV Cache
with a Progressive Compression Strategy", where attention weights and caches in the same layer are shared.

---

> ### Author Response · Authors · 2025-11-25
> **Response to Reviewer F8ju**
>
> Thank you for your constructive and valuable suggestions. We provide our responses as follows:
>
> ### **1. Analysis of Rank and the GQA architecture (corresponding to Weakness 1).**
>
> This is a valuable question. We provide additional experiments analyzing the SVD rank as a hyperparameter, as shown in the table below:
>
> | Method            | QA     | SUM    | FShot  | Synth  | Code   | Avg    | Ratio  |
> |------------------|--------|--------|--------|--------|--------|--------|--------|
> | baseline          | 17.20  | 27.85  | 68.31  | 39.87  | 61.15  | 42.88  | 1.000  |
> | rank==4096        | 19.33  | 24.37  | 65.36  | 43.17  | 57.78  | 42.00  | 0.714  |
> | rank==4096*0.8    | 18.75  | 24.67  | 62.48  | 43.92  | 58.02  | 41.57  | 0.571  |
> | rank==4096*0.7    | 19.88  | 24.59  | 64.62  | 41.40  | 57.87  | 41.67  | 0.500  |
> | rank==4096*0.6    | 19.74  | 23.84  | 60.71  | 44.00  | 55.15  | 40.69  | 0.429  |
> | rank==4096*0.5    | 15.54  | 22.35  | 58.05  | 42.67  | 52.83  | 38.29  | 0.357  |
>
> Since the primary compression benefit of the proposed method comes from cross-layer sharing rather than low-rank decomposition, we do not apply highly fine-grained rank configurations in the paper. Following Palu and assigning an independent rank to each layer may potentially yield further gains.
>
> For the GQA architecture, we consider the issue we encounter to be a common challenge shared by all low-rank decomposition methods.
> The dimensionality of the KV cache in GQA models is further reduced, making the benefits of single-layer SVD-based methods increasingly limited on such architectures.
> Our hypothesis is that grouping and sharing the KV cache is, to some extent, equivalent to how traditional low-rank methods compress the cache in MHA architectures, thereby reducing the effectiveness of such approaches.
> Building on this understanding, the proposed CommonKV mitigates the reliance on low-rank decomposition through cross-layer sharing, allowing it to achieve higher compression rates even under the GQA architecture.
>
> ### **2. Comparison with LORC (corresponding to Question1).**
>
> We apologize for the absence of discussion on LORC in the related work section.
> In our view, both LORC and Palu employ SVD-based compression applied to parameters within a single layer, which differs from the cross-layer sharing paradigm underlying the proposed method.
> The compression ratio achievable by single-layer compression is constrained by the SVD rank, which leads to substantial performance degradation under high compression settings (especially for GQA models).
> In contrast, cross-layer compression alleviates the rank constraints faced at the single-layer level and enables higher compression ratios by sharing the KV cache across different layers.
> We will add a discussion of LORC to the related work section.

---

### Official Review · Reviewer_ofNq · 2025-11-01

**Soundness:** 2
**Presentation:** 2
**Contribution:** 2
**Rating:** 2
**Confidence:** 3

**Summary:**

This paper proposes CommonKV. CommonKV is a KV cache compression method for long-context LLM inference that is training-free and leverages cross-layer compression. The key idea is to make adjacent layers' KV caches more mergeable by sharing part of the K/V projection parameters across nearby layers. To be more specific, this is done by performing a concatenated SVD over groups of layers, which yields a shared matrix and layer-specific matrices. Evaluation on major long-context benchmarks (LongBench and RULER) with 7/8B LLMs show that CommonKV preserves accuracy while enabling significant compression ratio.

**Strengths:**

Thank you for submitting this paper to ICLR! KV cache compression is one of the most important and popular topics in efficient LLM inference. I appreciate the authors' efforts on the analysis of limitations in existing methods (not training free, and performance degradation under aggressive compression). Cross-layer parameter sharing is a very reasonable idea (somehow explored before). The evaluation baselines are strong and timely methods in the field as well. In particular, section 6 gives a very comprehensive overview of methods in this area, which is a unique contribution beyond the proposed method itself.

**Weaknesses:**

1. As compared to other KV cache compression papers in the community, the evaluation is flawed in many aspects, including but not limited to context window length of 8K, model sizes (7/8B), rationale of hyperparameter choices, system performance metrics, etc. Please refer to "questions" for a comprehensive list.

2. There are quite a few unfilled question marks (??) and TODOs in the current draft.

3. Figures 1, 3, 4 are hard to read --- Please consider enlarging the fonts.

**Questions:**

Please justify the design choice of setting the context window of all models to be 8K. As an example, LongBench has an average input length of 6.7K and a lot of queries of length > 10K. It'd be great to see whether the proposed methods scales to longer contexts, especially given that KV cache compression could target ultra-long-context scenarios.

It is unclear whether the proposed approach scales to larger LLMs, especially those with > 8B parameters. Do you have intuitions on whether larger models with preserve the performance gain?

Could you give some sensitivity analysis of group size and SVD rank? I'm a bit confused by the current rationale of picking values for these two hyperparameters.

End-to-end inference latency is not a typical metrics that KV cache compression researchers use for system performance measurement. Please consider adding TTFT/throughput numbers.

---

> ### Author Response · Authors · 2025-11-25
> **Response to Reviewer ofNq (1/2)**
>
> Thank you for your constructive and valuable suggestions. We provide our responses as follows:
>
> ### **1. Explanation regarding the length of the evaluation data (corresponding to Weakness1 & Question 1).**
>
> We apologize for the confusion.
> Our decision to use 8k as the truncation length is based on the fact that the Llama3.1-8B-Instruct model natively supports an 8k context length according to its configuration.
> To ensure optimal model performance, we follow the official truncation strategy, which concatenates the prefix and suffix obtained by taking max_len // 2 from each end.
>
> | Method     | QA    | Sum    | FShot  | Synth  | Code   | Avg   |
> |------------|-------|--------|--------|--------|--------|-------|
> | fullkv     | 20.79 | 28.45  | 69.73  | 41.72  | 61.15  | 44.37 |
> | palu-0.5   | 19.74 | 26.07  | 66.42  | 27.58  | 36.54  | 35.27 |
> | ours-0.5   | 18.62 | 24.08  | 61.56  | 44.00  | 54.49  | 40.55 |
>
> To verify the effectiveness of the proposed method on longer inputs, we adjust the truncation length from 8k to 16k and conduct experiments accordingly (as you noted, this length can cover the vast majority of samples in LongBench).
> As shown in the table above, on the Llama-3.1-8B-Instruct model, CommonKV continues to exhibit a clear advantage over strong baseline methods when handling longer texts, demonstrating the proposed method’s generalization capability with respect to input length.
>
> ### **2. Correction of formatting issues (corresponding to Weakness 2 & 3).**
>
> We apologize for the inconvenience caused during your review due to our oversight. Several typographical errors are already corrected in the latest version. We appreciate your careful review and meaningful suggestions.
>
> ### **3. Additional experiments on larger models (corresponding to Question 2).**
>
> You raise an important point.
> Larger-scale models typically entail larger hidden states, which introduces greater challenges for cross-layer merging.
> Considering the high resource demands of long-text processing and our practical constraints, we choose the Llama2-13B-Instruct model for the experiments.
>
> | Table      | QA     | Sum    | FShot  | Synth  | Code   | Avg    |
> |------------|--------|--------|--------|--------|--------|--------|
> | fullkv     | 22.66  | 23.79  | 62.75  | 8.10   | 53.11  | 34.08  |
> | ours-0.6   | 25.91  | 22.73  | 62.50  | 6.90   | 46.49  | 32.91  |
> | palu-0.6   | 24.45  | 22.71  | 60.69  | 7.90   | 45.74  | 32.30  |
> | ours-0.3   | 24.31  | 21.65  | 61.61  | 11.00  | 42.53  | 32.22  |
> | palu-0.3   | 12.52  | 10.31  | 41.65  | 1.40   | 18.28  | 16.83  |
>
> As shown in the table above, the proposed method continues to maintain stable performance after KV cache compression on models exceeding 8B parameters.
> Under high compression ratios (retaining only 30% of the KV cache), it still demonstrates a substantial advantage over strong baselines Palu.

---

> ### Author Response · Authors · 2025-11-25
> **Response to Reviewer ofNq (2/2)**
>
> ### **4. Hyperparameter analysis of rank and group (corresponding to Question 3).**
>
> For methods such as Palu that perform compression within a single layer, the rank indeed serves as the key factor determining the compression ratio.
> In contrast, for CommonKV, the primary compression benefit derives from cross-layer merging, so the method does not place strong demands on the rank itself.
> For the SVD rank, we do not apply fine-grained tuning; instead, we adopt a uniform setting across all layers. With a group size of 4, the performance under different rank values is as follows:
>
> | Method            | QA     | SUM    | FShot  | Synth  | Code   | Avg    | Ratio  |
> |------------------|--------|--------|--------|--------|--------|--------|--------|
> | baseline          | 17.20  | 27.85  | 68.31  | 39.87  | 61.15  | 42.88  | 1.000  |
> | rank==4096        | 19.33  | 24.37  | 65.36  | 43.17  | 57.78  | 42.00  | 0.714  |
> | rank==4096*0.8    | 18.75  | 24.67  | 62.48  | 43.92  | 58.02  | 41.57  | 0.571  |
> | rank==4096*0.7    | 19.88  | 24.59  | 64.62  | 41.40  | 57.87  | 41.67  | 0.500  |
> | rank==4096*0.6    | 19.74  | 23.84  | 60.71  | 44.00  | 55.15  | 40.69  | 0.429  |
> | rank==4096*0.5    | 15.54  | 22.35  | 58.05  | 42.67  | 52.83  | 38.29  | 0.357  |
>
> As shown in the table above, the model’s performance under different rank settings leads us to select 0.7 as the final hyperparameter to balance performance and compression ratio.
> We do not conduct a more fine-grained search for the rank. If each layer is tuned more precisely (as in Palu), CommonKV should be able to achieve further performance gains.
>
> In addition, for the group-size hyperparameter, the range of viable choices is quite limited: excessively large groups (e.g., 8) introduce performance degradation, whereas overly small groups fail to provide meaningful parameter compression.
> We report the impact of group size on performance with the rank set to 0.7 × 4096.
> The experimental results are shown in the table below:
>
> | Method       | QA     | SUM    | FShot  | Synth  | Code   | Avg    | Ratio  |
> |-------------|--------|--------|--------|--------|--------|--------|--------|
> | baseline    | 17.20  | 27.85  | 68.31  | 39.87  | 61.15  | 42.88  | 1.000  |
> | merge==2    | 20.05  | 25.42  | 67.14  | 44.38  | 55.25  | 42.44  | 1.000  |
> | merge==4    | 19.88  | 24.59  | 64.62  | 41.40  | 57.87  | 41.67  | 0.500  |
>
> When the merge group is set to 2, due to the GQA-related reasons discussed in Sec. 4.1, there is not even any storage saving.
> With group = 4, the proposed method achieves a 50% compression ratio without noticeable performance degradation.
> This also indirectly indicates that CommonKV is not sensitive to the rank; the primary compression gains originate from cross-layer merging.
>
> ### **5. Selection of inference latency metrics (corresponding to Question 4).**
>
> Thank you for the suggestion. We adopt the more standard metric (TTFT and throughput) to report the latency results again, as shown in the table below.
>
> | Method   |  TTFT (s) | Throughput (tokens/s) |
> |----------|-----------------|----------------------|
> | xkv      | 15.69           | 35.59                |
> | ours     | 2.33            | 33.11                |
> | fullkv   | 2.16            | 34.72                |
> | palu     | 2.18            | 33.67                |
>
> It can be observed that, compared with xKV, our method introduces no noticeable overhead in TTFT during the prefill stage, and during decoding, compressing 50% of the KV cache does not lead to significant inference latency.
>
>
> We once again thank you for your careful and patient review, and we hope that our responses address your concerns.

---

> ### Author Response · Authors · 2025-11-28
> **Look Forward to More Feedback**
>
> Dear Reviewer ofNq,
>
> We sincerely appreciate your thoughtful and constructive feedback. We have diligently addressed each of your concerns in our point-by-point rebuttal.
>
> We hope our response has alleviated your concerns and illuminated the value of our work. If so, we would be immensely grateful if you could reconsider your recommendation. We assure you that all your insightful comments will be meticulously incorporated into the final manuscript. We have earnestly integrated feedback from all four reviewers and hope this is evident in your evaluation.
>
> Thank you for dedicating your valuable time to review our response.
>
> With deepest gratitude,
>
> Paper 16049 Authors

---

### Author Response · Authors · 2025-11-28
**General Response**

**Dear Reviewers, Area Chairs, Senior Area Chairs and Program Chairs,**

We sincerely thank all reviewers for their positive feedback and constructive comments. We are encouraged that the reviewers positively acknowledge the novelty of our cross-layer parameter sharing insight, the practicality of the training-free design, the impressive performance on benchmarks, and the comprehensive analysis provided in the paper. Specifically, Reviewers F8ju and 27QT highlight that our framework addresses the root causes of performance degradation in KV cache compression, while Reviewers ofNq and kcrW commend the thorough evaluation and orthogonality to existing methods:

**[Novelty & Insight]:**

- **Reviewer F8ju:** "The method's core strength is its novel idea of creating a more consistent latent cache via parameter sharing, which directly addresses the root cause of poor performance in direct KV cache merging."
- **Reviewer 27QT:** "Neat insight into using parameter sharing across layers to compress based on similarity between adjacent layers of the KV Cache."
- **Reviewer kcrW:** "An interesting observation regarding the KV-generator being compressible is utilized to effectively reduce inference overhead."

**[Practicality (Training-Free)]:**

- **Reviewer F8ju:** "Its training-free nature makes it highly practical and easy to apply to existing models."
- **Reviewer 27QT:** "Training-free method of KV merging leads to lower offline compute overhead."
- **Reviewer kcrW:** "No pretraining/fine-tuning makes it quite practical."

**[Performance & Efficacy]:**

- **Reviewer 27QT:** "Results are very impressive compared to all the baselines."
- **Reviewer F8ju:** "CommonKV demonstrates superior empirical performance over baselines at high compression ratios with minimal inference latency overhead."
- **Reviewer kcrW:** "Strong performance with good latency analysis."

**[Orthogonality & Comprehensive Analysis]:**

- **Reviewer ofNq:** "Section 6 gives a very comprehensive overview of methods in this area, which is a unique contribution beyond the proposed method itself."
- **Reviewer F8ju:** "Its ability to be combined with quantization and eviction methods to achieve up to 98% compression is a significant advantage."
- **Reviewer kcrW:** "Also orthogonal to other optimizations (eviction/quantization)."

In the past weeks, we have carefully addressed the specific questions and requests raised by the reviewers to further strengthen the paper. We hope our responses and the revised manuscript satisfactorily address all concerns. Once again, we would like to express our sincere appreciation to the Area Chair and Reviewers for their engagement and time invested in this process.

Best regards,

Authors of **CommonKV**

---

### Note · Authors · 2025-12-01

I have read and agree with the venue's withdrawal policy on behalf of myself and my co-authors.